# A *TAC3* Missense Variant in a Domestic Shorthair Cat with Testicular Hypoplasia and Persistent Primary Dentition

**DOI:** 10.3390/genes10100806

**Published:** 2019-10-14

**Authors:** Petra Hug, Patricia Kern, Vidhya Jagannathan, Tosso Leeb

**Affiliations:** 1Institute of Genetics, Vetsuisse Faculty, University of Bern, 3001 Bern, Switzerland; petra.hug@vetsuisse.unibe.ch (P.H.); vidhya.jagannathan@vetsuisse.unibe.ch (V.J.); 2Tierarztpraxis Spiegelberg AG, 4566 Halten, Switzerland; info@tierarztpraxis-spiegelberg.ch

**Keywords:** *Felis catus* L., whole genome sequence, animal model, neurokinin B, puberty, development, precision medicine, disorder of sexual development

## Abstract

A single male domestic shorthair cat that did not complete puberty was reported. At four years of age, it still had primary dentition, testicular hypoplasia, and was relatively small for its age. We hypothesized that the phenotype might have been due to an inherited form of hypogonadotropic hypogonadism (HH). We sequenced the genome of the affected cat and compared the data to 38 genomes from control cats. A search for private variants in 40 candidate genes associated with human HH revealed a single protein-changing variant in the affected cat. It was located in the *TAC3* gene encoding tachykinin 3, a precursor protein of the signaling molecule neurokinin B, which is known to play a role in sexual development. *TAC3* variants have been reported in human patients with HH. The identified feline variant, *TAC3*:c.220G>A or p.(Val74Met), affects a moderately conserved region of the precursor protein, 11 residues away from the mature neurokinin B sequence. The affected cat was homozygous for the mutant allele. In a cohort of 171 randomly sampled cats, 169 were homozygous for the wildtype allele and 2 were heterozygous. These data tentatively suggest that the identified *TAC3* variant might have caused the suppression of puberty in the affected cat.

## 1. Introduction

Gonadotropin releasing hormone (GnRH) is a main regulator of the reproductive endocrine system. Its secretion determines the pattern of secretion of luteinizing hormone (LH) and follicle-stimulating hormone (FSH). LH activates the release of testosterone in Leydig cells in the testicle. Testosterone is responsible for the male phenotype, body growth, and sperm production [1]. Coordinated and pulsatile GnRH secretion is induced due to the synergistic action of neurokinin B, kisspeptin, and dynorphin [2]. Neurokinin B stimulates kisspeptin neurons, which leads to GnRH secretion [3]. Dysfunctional GnRH release leads to low blood testosterone and pituitary hormone levels, resulting in hypogonadotropic hypogonadism (HH) [4]. Impaired testicular function may occur due to a primary testicular disorder or secondary to hypothalamic–pituitary dysfunction (hypogonadotropic). HH may be inherited or caused by non-genetic factors, such as drugs, encephalic trauma, or infiltrative or inflammatory pituitary lesions. Congenital hypogonadotropic hypogonadism (cHH) is a rare genetic disorder characterized by a delayed or absent pubertal development, micropenis, cryptorchidism, androgen/estrogen deficiency, and infertility due to an inadequate secretion of gonadotrophin-releasing hormone (GnRH), with an otherwise structurally and functionally normal hypothalamus and pituitary gland. Low testosterone and pituitary hormone levels confirm the diagnosis [4]. Variants in 40 genes have been shown to cause different inherited forms of HH in humans (Table 1).

This study was initiated after an owner reported a male cat with missing puberty characteristics, such as normal testicle growth and change in primary dentition. The goal of this study was to identify a possible underlying causative genetic defect.

## 2. Materials and Methods 

### 2.1. Ethics Statement

All animal experiments were performed according to local regulations. The cat in this study is privately owned and was examined with the consent of the owner. The “Cantonal Committee for Animal Experiments” approved the collection of blood samples (Canton of Bern; permit 75/16).

### 2.2. Animal Selection

A 3-year-old male domestic shorthair cat with testicular hypoplasia and a persistent primary dentition was investigated. An EDTA blood sample was collected for genomic DNA isolation. Additionally, we used 171 blood samples from cats of various breeds which had been donated to the Vetsuisse Biobank (Appendix A). These samples represented population controls without reports of testicular hypoplasia or a persistent primary dentition.

### 2.3. Hormone Measurement

The testosterone level of the affected cat was measured from an EDTA blood sample by labor-zentral.ch (Geuensee, Switzerland).

### 2.4. DNA Extraction

Genomic DNA was isolated from EDTA blood with the Maxwell RSC Whole Blood Kit using a Maxwell RSC instrument (Promega). Additionally we used DNA from EDTA blood of 171 non-affected control cats of various breeds that had been stored in our biobank.

### 2.5. Whole Genome Sequencing

An Illumina TruSeq PCR-free DNA library with 350 bp insert size of the affected cat (K532) was prepared. We collected 168 million 2 × 150 bp paired-end reads on a NovaSeq 6000 instrument (18x coverage). Mapping and alignment were performed as described [35]. The sequence data were deposited under the study accession PRJEB7401 and the sample accession SAMEA5885924 at the European Nucleotide Archive.

### 2.6. Variant Calling

Variant calling and filtering was performed as described [35]. To predict the functional effects of the called variants, SnpEFF [36] software, together with NCBI annotation release 104 for the Felis_catus_9.0 assembly, was used. For variant filtering we used 38 control genomes, which were produced during other projects of our group (Appendix A).

### 2.7. Gene Analysis

We used the Felis_catus_9.0 cat reference genome assembly for all analyses. Numbering within the feline *TAC3* gene corresponded to the NCBI RefSeq accessions XM_003988924.5 (mRNA) and XP_003988973.1 (protein).

### 2.8. Sanger Sequencing

The *TAC3*:c.220G>A variant was genotyped by direct Sanger sequencing of PCR amplicons. A 400 bp PCR product was amplified from genomic DNA using AmpliTaqGold360Mastermix together with primers 5‘-AGC CCA CTT CTC TTC CAG TG -3‘ (Primer F) and 5’-AGA GGG GAT TCA GGT CAC AA-3’ (Primer R). After treatment with exonuclease I and alkaline phosphatase, amplicons were sequenced on an ABI 3730 DNA Analyzer. Sanger sequences were analyzed using Sequencher 5.1 software.

## 3. Results

### 3.1. Clinical Examination

A 3-year-old male domestic shorthair cat was presented with persistent primary dentition consisting of one primary maxillary canine (Figure 1). Upon examination there was one small right testicle located in the scrotum. The left testicle could not be located. It was neither scrotal, nor palpable in the inguinal area. The external genitalia, including the urethral orifice, were in the normal position, although with a juvenile appearance because of their small size. The hair coat had an unkempt appearance. The cat had small body size but proportional growth. It had reportedly displayed mounting behavior toward another female cat in the household and showed an increasingly dominant–aggressive behavior toward other cats outside. The cat was presented for a follow-up examination at 4 years of age. No changes in behavior or the stage of adolescence were noticed.

### 3.2. Laboratory Findings

The blood testosterone level of the cat at the age of 4 years was 0.28 ng/mL, which was below the reference range for cats (0.92 ng/mL–9.17 ng/mL).

### 3.3. Genetic Analysis

We sequenced the genome of the affected cat and searched for homozygous and heterozygous variants in known candidate genes that were not present in 38 control cats of different breeds (Table 2).

This analysis identified a single homozygous private protein-changing variant in *TAC3*, a known HH candidate gene [32]. The variant was designated as ChrB4:85,517,451C>T (Felis_catus_9.0 assembly). It is a missense variant, XM_003988924.5:c.220G>A, predicted to result in the amino acid change XP_003988973.1:p.(Val74Met). This amino acid residue is moderately conserved across mammals. Primates, including humans, have a threonine at this position, while all other investigated mammalian TAC3 sequences have a valine (Figure 2). In silico predictions of the functional effect yielded conflicting results. PolyPhen-2 predicted the variant as benign with a score of 0.147 [37]. SIFT predicted that the variant affected protein function, with a score of 0.03 [38]. The SIFT prediction had low confidence due to the limited diversity of the available related sequences.

We confirmed the presence of the *TAC3* variant by Sanger sequencing and genotyped 171 control cats (Figure 3). The case was homozygous for the mutant allele. Two domestic shorthair cats without reports of testicular hypoplasia or persistent primary dentition carried the alternative allele in a heterozygous state (Table 3; Appendix A).

## 4. Discussion

In this study, we identified a *TAC3*:c.220G>A (p.Val74Met) missense variant in a domestic shorthair cat with absent puberty characteristics, such as normal testicle growth and change in primary dentition. The phenotype closely resembled HH, which belongs to the larger group of disorders of sexual development (DSD). The predicted amino acid substitution affects a moderately conserved region of the TAC3 precursor protein, which is 11 residues away from the mature neurokinin B sequence. The affected cat was homozygous for the mutant allele, which was found to be rare in the normal cat population. It should also be noted that the variant is close to the exon/intron boundary and might have an effect on splicing. Due to a lack of suitable RNA samples, we could not experimentally assess the *TAC3* mRNA splicing pattern in the affected cat.

Cases of human HH have been reported to be caused by variants in either the *TACR3* gene encoding the neurokinin B receptor or the *TAC3* gene itself [32,39,40,41,42,43]. The phenotype of these patients is termed normosmic congenital hypogonadotropic hypogonadism (ncHH; OMIM #614839). ncHH is clinically characterized by a failure to enter puberty. Female patients have amenorrhea, absent breast development, and hypoplastic ovaries and uteri. Male patients have small testicles and micropenis. Body hair growth in ncHH patients resembles a juvenile pre-puberty state. ncHH in humans can be successfully treated by pulsatile administration of GnRH [42] or steroid hormones [43]. The known human pathogenic *TAC3* variants consist of frameshift [43] and splice site [41,42] variants and a single missense variant, p.M90T, which affects the C-terminal residue of the mature neurokinin B peptide [32]. Additional likely pathogenic missense variants affecting the mature neurokinin B sequence were deposited in the ClinVar database.

In felines, efforts are underway to develop a method to permanently sterilize cats by RNAi-mediated silencing of *KISS1* and *TAC3*. This method is predicted to lead to a reduction in the stray animal population and therefore decrease animal suffering and vectors for human disease [44].

## 5. Conclusions

Our genetic data and existing knowledge regarding the physiological function of *TAC3* suggest that the *TAC3*:c.220G>A (p.Val74Met) missense variant may be considered a candidate causative variant for the observed HH phenotype in the studied cat. Given that this is a single case investigation and that we have no functional confirmation of neurokinin B deficiency, this result must be considered preliminary and should be interpreted with caution.

## Figures and Tables

**Figure 1 genes-10-00806-f001:**
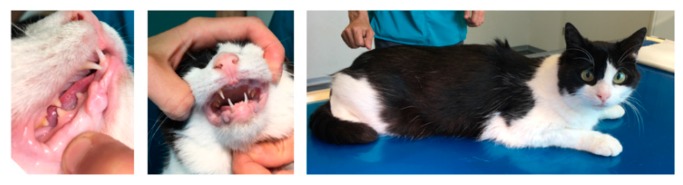
Clinical phenotype characterized by persistent primary dentition and an “unkempt” coat appearance.

**Figure 2 genes-10-00806-f002:**
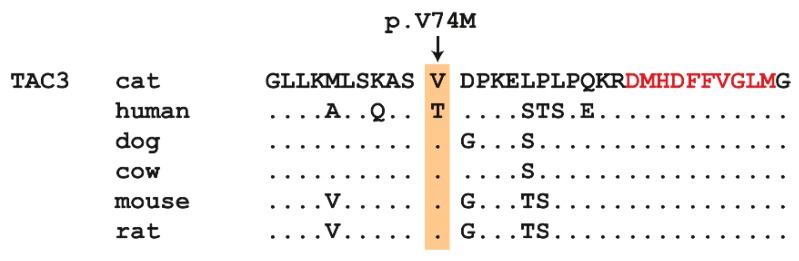
Multi-species protein alignment of the TAC3 precursor protein in the region of the missense variant. The valine at position 74 of the feline TAC3 protein is conserved in most mammals, except primates. The 10 amino acid sequence of the mature neurokinin B signaling peptide is indicated in red. Amino acid sequences were derived from XP_003988973.1 (cat); NP_037383.1 (human); XP_005625588.1 (dog); NP_851360.1 (cattle); NP_033338.2 (mouse); NP_062035.1 (rat).

**Figure 3 genes-10-00806-f003:**
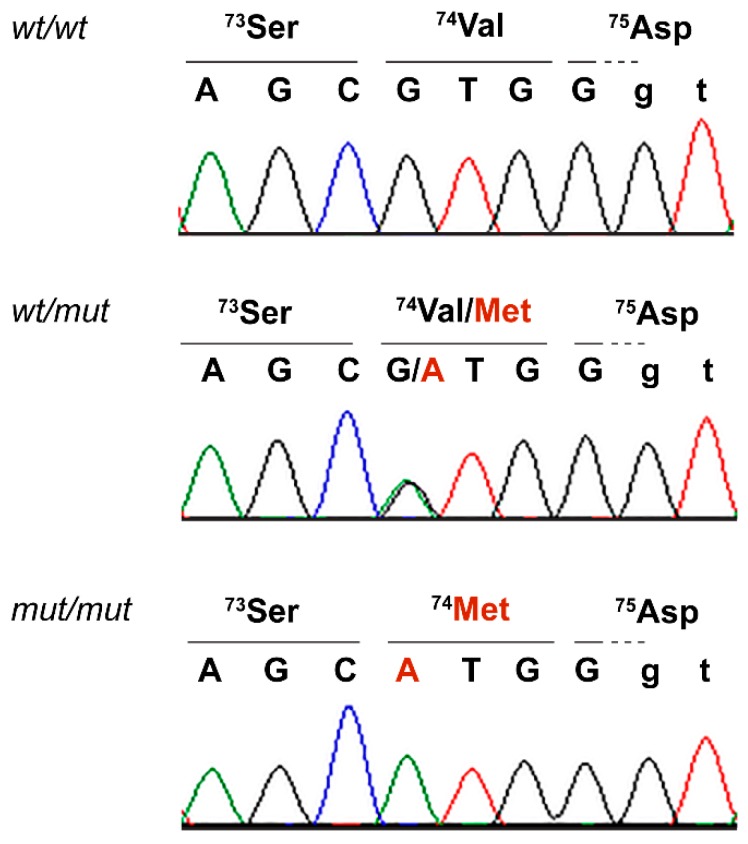
Details of the *TAC3*:c.220G>A variant. Representative electropherograms of three cats with different genotypes are shown. Exonic bases are shown in capital letters and intronic bases in lowercase letters.

**Table 1 genes-10-00806-t001:** Overview on genetic causes of human HH.

Gene	Phenotype	Inheritance	Ref.
*CHD7*	Hypogonadotropic hypogonadism 5 with or without anosmia	AD	[5]
*DAX1*	Adrenal hypoplasia, congenital	XLR	[6]
*DUSP6*	Hypogonadotropic hypogonadism 19 with or without anosmia	AD	[7]
*FEZF1*	Hypogonadotropic hypogonadism 22 with or without anosmia	AR	[8]
*FGF8*	Hypogonadotropic hypogonadism 6 with or without anosmia	AD	[9]
*FGF17*	Hypogonadotropic hypogonadism 20 with or without anosmia	AD	[7]
*FGFR1*	Hypogonadotropic hypogonadism 2 with or without anosmia	AD	[10]
*FLRT3*	Hypogonadotropic hypogonadism 21 with anosmia	AD	[7]
*FSHB*	Hypogonadotropic hypogonadism 24 without anosmia	AR	[11]
*GNRH1*	Hypogonadotropic hypogonadism 12 with or without anosmia	AR	[12]
*GNRHR*	Hypogonadotropic hypogonadism 7 without anosmia	AR	[13]
*HESX-1*	Growth hormone deficiency with pituitary anomalies	AR, AD	[14]
*HS6ST1*	Hypogonadotropic hypogonadism 15 with or without anosmia	AD	[7]
*IL17RD*	Hypogonadotropic hypogonadism 18 with or without anosmia	AR, AD	[7]
*KAL1*	Hypogonadotropic hypogonadism 1 with or without anosmia	XLR	[15]
*KISS1*	Hypogonadotropic hypogonadism 13 with or without anosmia	AR	[16]
*KISS1R*	Hypogonadotropic hypogonadism 8 with or without anosmia	AR	[17]
*LEP*	Obesity, morbid, due to leptin deficiency	AR	[18]
*LEPR*	Obesity, morbid, due to leptin deficiency	AR	[18]
*LHB*	Hypogonadotropic hypogonadism 23 with or without anosmia	AR	[19]
*LHX3*	Pituitary hormone deficiency, combined, 3	AR	[20]
*NELF*	Hypogonadotropic hypogonadism 9 with or without anosmia	AD	[21]
*OTUD4*	Gordon Holmes syndrome	AR	[22]
*PNPLA6*	Boucher–Neuhauser Syndrome	AR	[23]
*POLR3B*	hypogonadotropic hypogonadism	AR	[24]
*PROK2*	Hypogonadotropic hypogonadism 4 with or without anosmia	AD	[25]
*PROKR2*	Hypogonadotropic hypogonadism 3 with or without anosmia	AD	[25]
*PROP-1*	Pituitary hormone deficiency, combined, 2	AR	[26]
*RAB18*	Warburg micro syndrome 3	AR	[27]
*RAB3GAP1*	Warburg micro syndrome 1	AR	[27]
*RAB3GAP2*	Warburg micro syndrome 2	AR	[27]
*RNF216*	Cerebellar ataxia and hypogonadotropic hypogonadism	AR	[28]
*SEMA3A*	Hypogonadotropic hypogonadism 16 with or without anosmia	AD	[29]
*SOX2*	Abnormalities of the central nervous system	AD	[30]
*SPRY4*	Hypogonadotropic hypogonadism 17 with or without anosmia	AD	[7]
*STUB1*	Spinocerebellar ataxia, autosomal recessive 16	AR	[31]
*TAC3*	Hypogonadotropic hypogonadism 10 with or without anosmia	AR	[32]
*TACR3*	Hypogonadotropic hypogonadism 11 with or without anosmia	AR	[32]
*TBC1D20*	Warburg micro syndrome 4	AR	[33]
*WDR11*	Hypogonadotropic hypogonadism 14 with or without anosmia	AD	[34]

**Table 2 genes-10-00806-t002:** Results of variant filtering.

Filtering Step	Homozygous Variants	Heterozygous Variant
Private variants	25,355	209,967
Protein-changing private variants	111	756
Private variants in known candidate genes	1	0

**Table 3 genes-10-00806-t003:** Genotype phenotype association of the *TAC3*:c.220G>A variant.

Cats	G/G	G/A	A/A
Cases (*n* = 1)	0	0	1
Control cats (*n* = 171)	169	2	0

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
