# Peer review of "A TAC3 Missense Variant in a Domestic Shorthair Cat with Testicular Hypoplasia and Persistent Primary Dentition"

_genes, 2019, doi:10.3390/genes10100806_

Round 1
Reviewer 1 Report
In the manuscript a novel gene mutation in a cat with abnormal phenotype, tentatively classified as hypogonadotropic hypogonadism, was reported. Whole genome sequencing of the patient was performed and compared with 38 available whole genome sequences of 38 control cats. The molecular analysis is convincing and very well documented.
Remarks and suggestions:
1. Description of the phenotype (unilateral cryptorchidism, underdeveloped testis, low level of testosterone) and the role of TAC3 gene indicate that disorder of sex development (DSD) in the patient also was observed. Thus, the following issues should be clarified:
a) appearance of external genitalia, including position of the urethral orifice,
b) USG inspection of the abdomen should be performed (searching for female reproductive tracts?)
c) cytogenetic and molecular (e.g. SRY gene) detection of sex chromosomes should be performed to confirm male sex of the patient
2. Discussion
a) more information on human patients with TAC3 mutations (position of the mutation with the gene, phenotype) should be given,
b) issue of DSD phenotype should be shortly discussed.
Author Response
Please see the attached cover letter.

Reviewer 2 Report
The paper is well written and the experimental design/validation is adequate.
The conclusions are conservative but appropriate based on the data presented.
Minor edits
Figure 3 the top panel indicates glycine for amino acid 75 instead of glutamine.
Line (119) the Sift output for sift analysis indicates that the protein will be affected with the stated score. However, in my analysis with the online program, the output also indicates the prediction has low confidence given the diversity of the sequences used. If this is the case, then it must be stated, if not the additional protein sequences the negate this qualification used should be included.
Consideration
The identified variant occurs 4 nucleotides from the donor splice site of exon 2. The predicted putative donor acceptor site GG-AT is extremely rare, representing non-canonical mammalian splice site found in less than 0.02% splice junctions. If the putative boundary of exon 2 is shortened by two bases and final two nt of intron 2 are included in exon 3, the putative donor acceptor site is the canonical GT-AG found in over 99% of mammalian splice sites. The predicted amino acid sequence does not change with the 2 nt shift. However, the identified variant changes from a valine to an isoleucine as opposed to methionine. Sift analysis and predict SNP both suggest the isoleucine mutation would be tolerated/neutral. However, if the 2 nt shift is accurate the identified mutation disrupts the GT-AG donor acceptor site changing to AT-AG (again found in less than 0.02%). In the absence of sequencing the transcript, this is hypothetical but since the transcript and exon/intron structure is also predicted, it too is hypothetical. This possibility should be considered for inclusion in the discussion.
Author Response
Please see the attached cover letter.
